# Characterization of Extended-Spectrum Beta-Lactamase-Producing *Escherichia coli* Isolates from Jurong Lake, Singapore with Whole-Genome-Sequencing

**DOI:** 10.3390/ijerph18030937

**Published:** 2021-01-22

**Authors:** Yang Zhong, Siyao Guo, Kelyn Lee Ghee Seow, Glendon Ong Hong Ming, Joergen Schlundt

**Affiliations:** 1Nanyang Technological University Food Technology Centre (NAFTEC), 62 Nanyang Drive, Singapore 637459, Singapore; SGUO004@e.ntu.edu.sg (S.G.); kseow@ntu.edu.sg (K.L.G.S.); hongming001@e.ntu.edu.sg (G.O.H.M.); 2School of Chemical and Biomedical Engineering, Nanyang Technological University, 62 Nanyang Drive, Singapore 637459, Singapore

**Keywords:** ESBL, Singapore, mcr-1

## Abstract

*Background*: The fast-spreading of Extended-Spectrum Beta-Lactamase-Producing *Escherichia coli* (ESBL-producing *E. coli*) and ESBL genes has become a big challenge to public health. The risk of spreading ESBL genes and pathogens in the environment and community has raised public health concern. The characterizing and whole-genome sequencing studies of ESBL-producing bacteria from reservoir water in Singapore is still limited. *Materials and methods*: The reservoir water sample was taken from two randomly selected sampling points of the Chinese Garden (Jurong river reservoir), which is a popular reservoir park in Singapore. The bacteria of the water sample were collected with 0.45 µm filter membranes and enriched before processing with ESBL-producing *E. coli* screening. The collected ESBL positive isolates were further characterized by both phenotypic tests including disc diffusion and microdilution Minimum Inhibitory Concentration (MIC) test, and also genotypic test as whole-genome sequencing analysis. Besides, to investigate the transferability of the resistance gene, a conjugation test was performed with the J53 *E. coli* strain as the gene receptor. *Result:* Nine ESBL-producing *E. coli* isolates were collected and confirmed as ESBL-producing with both phenotypic and genotypic tests. A potential pathogen as ST131 clade A isolate was identified, and all isolates were determined to harbor a *bla*_CTX-M_ gene. Among them, strain J1E4 was resistant to polymyxin E and confirmed to harboring a conjugatable mcr-1 gene. Further genetic environment analysis has reflected a conversed gene cluster formed by insert sequence (IS), *bla*_CTX-M-15_, and *WbuC* family cupin-fold metalloprotein, which may potentially jump from the plasmids to the chromosome. *Conclusion*: The first time we reported the whole genome sequencing (WGS) data of ESBL-producing *E. coli* including potential pathogen (ST131) present in reservoir water in Singapore. The ESBL-producing *E. coli* from reservoir water also carrying conjugatable colistin resistance genes which may become a risk to human health.

## 1. Introduction

Extended-Spectrum Beta-Lactamase (ESBL)-producing Enterobacteriaceae has become a big challenge to infection control due to their resistance to most of the beta-lactams. The ESBL can hydrolyze the most commonly used penicillin, monobactam, and cephalosporin including the 3rd generation [1]. Moreover, ESBL-producing bacteria are frequently reported to carry multi-drug resistance (MDR), which limits the options of antimicrobials to treat infections caused by them [2]. The prevalence of ESBL-*E. coli* is increasing fast worldwide, especially some prevalent sequencing types like ST131. A study from Europe in 2018 has reported a fraction of 20.5% of ST131 among the ESBL-producing *E. coli* from four European hospitals [3]. As the situation gains importance, a higher requirement of sequencing information for deeper genetic alignment of the ESBL-producing *E. coli* isolates has been addressed. Besides the Multilocus sequence typing (MLST) with the polymerase chain reaction (PCR) and Sanger sequencing, the whole genome sequencing (WGS) has been increasingly applied in the phylogenetic analysis of ESBL-producing *E. coli* isolates. Further WGS analysis helps to classify ST131 isolates into different clades and makes it easier for global researchers to monitor the dissemination of them [4].

Nowadays, besides the spreading of ESBL-producing *E. coli* isolates, the horizontal transfer of beta-lactamase genes between different sources and species has also become a big concern. [5] Besides traditional phenotypic testing, genetic methods like PCR and Sanger sequencing targeting ESBL genes have made great contributions in monitoring the spread of ESBL genes [6]. Moreover, the sequencing technology and the Antimicrobial Resistance (AMR) gene database have been widely applied in this field to promote the study of the location and combination of the AMR gene cluster within relevant isolates. Surveillance systems from many countries have been upgraded to include the ESBL genes found in different types of isolates. Among them, the *bla*_CTX-M_ genes are one of the most prevalent ESBL genes. As addressed by the Centers for Disease Control and Prevention (CDC),

(https://www.cdc.gov/drugresistance/biggest-threats.html#extend), the combination of the *bla*_CTX-M_ gene and ST131 has great clinical importance.

Surface water plays an important role in providing reservoirs for microorganisms to exchange their genetic information including AMR markers. ESBL-producing bacteria have been widely reported in water bodies in many countries [7,8]. Furthermore, with the help of WGS, the genetic studies of the relationship among resistant isolates from different water sources in the same region have shown interesting new trends as per the spread of these bacteria [9]. 

The reservoirs and the rainwater catchment systems in Singapore are very unique and play an important role in freshwater supply. To avoid pollution, the wastewater collection system is separated from the rainwater collection. However, the reservoir water may still have the potential to obtain AMR markers like ESBL genes originating in other environmental and anthropogenic sources, such as soil, air, plants, animals, and human beings [10]. The Jurong Lake reservoir is located in a 90-hectare new national garden in western Singapore and is planning to build the largest commercial and regional center outside Singapore’s city center. It is available for water sports, popular for community activities. The water samples were taken from two points of the reservoir to collect ESBL-producing *E. coli* strain potential contaminated from the community. Based on our knowledge, our study is the first WGS analysis study of ESBL-producing *E. coli* from the reservoir water in Singapore. This study aims towards a genetic characterization of ESBL-producing *E. coli* from reservoir water, further, investigate their phylogenetic relationship with isolates from humans, report the acquired ESBL genes as well as other resistance genes harbored by the isolates, predict the transferability of the AMR genes with sequencing data. With the reported data, we have provided some new insights for discussing the potential role of this water environment in the spread of clinically relevant AMR in Singapore.

## 2. Materials and Methods

### 2.1. Sample Collection and Processing

Eight hundred milliliters of surface water (1 m depth to the surface) was collected from two randomly selected sampling points in Chines Garden and transported to the lab at 4 °C within 4 h. The full water sample was then filtered through 0.45 µm filter membranes (Sigma, Germany). The filter membranes were incubated overnight in Mueller Hinton (MH) Broth under 37 °C.

### 2.2. ESBL-Producing Bacteria Isolation

The overnight cultures were streaked on chromogenic screening plates as Brilliance ESBL agar plates (Thermo scientific, USA). Blue or pink colonies were picked as presumptive positive for *E. coli* and re-streaked on an MH agar for further purification. The pure colonies were picked for overnight liquid culture in MH broth at 37 °C. The overnight culture was then modified to long-term stock containing 20% (*v/v*) of glycerol and stored at −80 °C for further processing.

### 2.3. Antimicrobial Susceptibility Test

The ESBL screening was performed with a double-disc synergy test. Discs of amoxicillin/clavulanic acid (AMC, 30 µg) were placed adjacent to four cephalosporin discs including ceftriaxone (CRO, 30 µg), ceftazidime (CAZ, 30 µg), cefotaxime (CTX, 30 µg), and cefepime (FEP, 30 µg) with an 8-channel dispenser (Thermo Scientific, USA). The result was considered positive when elliptical clearing was apparent between the AMC disc and any of the cephalosporin discs. 

The ESBL confirmation was processed using Sensititre™ ESBL Plate (Thermo Scientific, USA) with microdilution methods. The details of the antimicrobials are listed in Appendix A. The Minimum Inhibitory Concentration (MIC) was defined as the lowest concentration to inhibit visible growth. The isolates were considered as ESBL-producing when an 8 times decrease of MICs appeared when cephalosporin combined with clavulanate acid compared to cephalosporin alone. 

### 2.4. DNA extraction and Next-Generation Sequencing

The bacterial genomic DNA was extracted using the Qiagen DNA miniprep kit (Germany) with the modified protocol [11]. The genomic DNA was then processed with Hiseq sequencing (Illumina, USA) based on the methods described before [12].

### 2.5. Whole-Genome Sequencing Analysis

The raw reads from Hiseq were assembled with Assembler1.2 [13]. The sequencing analysis tools used in this research is provided by the Center for Genomic Epidemiology (CGE) server [14]. The default setting was used for all tools. The genome annotation was performed with RAST [15]. The location of the beta-lactamase genes, fluoroquinolone resistance genes, and *mcr* genes were determined by the alignment of contigs carrying AMR genes with BLASTn. For isolates J1E1, J2E2, and J2E4, as the contigs carrying AMR genes were over the size limit of BLASTn, 20,000 bp fragments containing the AMR genes were extracted for Blast. The insert sequence was detected with ISFinder [16].

### 2.6. Phylogenetic Analysis

Multilocus sequence typing was processed with MLST 2.0 CGE server. As strain J2E1 has been defined to be ST131, a pack of ST131 isolate from different countries from the clinic was chosen to do the phylogenetic analysis with J2E1. The WGS raw reads of clinical *E. coli* isolates were selected from Bioproject accession No.PRJNA398288, as clinical isolates of ST131 from the year 2015 to 2017; the isolates details can be found in Appendix A. The phylogenetic analysis was processed with the CGE server based on the core gene single-nucleotide polymorphism (SNP). The phylogenetic tree was annotated with iToL tools [17]. The alignment of contigs carrying beta-lactamase genes was processed with BLASTn and plotted with SnapGene software (from GSL Biotech; available at snapgene.com). 

### 2.7. Conjugation Experiment to Test the Transferability of the Mcr-1 Gene

As harboring the *mcr-1* gene, strain J1E4 was chosen as the donor strain for the conjugation test. *E. coli* strain J53 with sodium azide resistance was chosen as recipients. Briefly, single colonies of both J1E4 and J53 was picked from overnight culture MH agar plate, respectively, and cultured in MH broth at 37 °C with 150 rpm for 4 h to reach the early log phase. The donor and recipient culture were mixed in a 3:1 (V: V) ratio and cultured at 37 °C overnight statically. The MH agars contain 200 µg/mL sodium azide was used to calculate the total concentration of the J53 strain. The MH agars contain 200 µg/mL sodium azide and 4 µg/mL colistin were used to select the J53 harboring *mcr-1* transconjugants. The antimicrobial susceptibility of successful transconjugants was determined with the microdilution MIC test and disc diffusion test. The co-transferability of *bla*_CTX-M-15_, *bla*_TEM-1B_ genes with the *mcr-1* gene was determined with the PCR test, the primers are shown as Appendix A, and the result was checked with agarose (2%, *w/v*) gel electrophoresis.

## 3. Results

### 3.1. Genome Profile of ESBL-Producing E. coli Isolates

Nine ESBL-producing *E. coli* were isolated from the water sample and sent for WGS. The basic genomic and typing information is listed in Table 1. The genome size of the nine isolates is between 4,866,444 bp to 5,408,697 bp and content GC percentage between 50.3% to 50.7%. The MLST types of isolates were all different. Two isolates had up to 5 plasmid replicons, three isolates had 2 origins of replications (Ori), the last four isolates carried 3–4 Oris. The most common Ori was *IncFII*, which was found in six isolates. The MLST types and plasmids are listed in Table 1. Among them, ST131 has been reported to cause infections in Singapore before and clinic isolates information is shown in Appendix A and phylogenetic tree.

### 3.2. AMR Genes of the Isolates

The AMR genes detected from each isolate are listed in Table 2 and shown in Figure 1. All the nine isolates shown positive ESBL in double-disc diffusion were confirmed to carrying ESBL genes. Besides, 8/9 except J1E2 harboring resistance genes subject to at least three classes, which from the genetic angle predict as MDR. As carrying the most AMR genes, strain J2E3 had been detected to carry 17 resistance genes subject to nine types of antimicrobials. All isolates were carrying *bla*_CTX-M_ genes, and two of the isolates were carrying a *bla*_TEM-1B_ gene besides the *bla*_CTX-M-15_. *bla*_CTX-M-15_ was the most prevalent beta-lactamase as it was detected in 5 isolates. Besides the beta-lactamase genes, the macrolide, lincosamide, and streptogramin B (MLS) classes of resistance genes were the most common and were detected in all isolates. Moreover, the variant type is mainly *mph(A)* and *mdf(A).*


Fluoroquinolone resistance gene *qnrS1* was also detected in four isolates including a variant strain of *qnrS1* with 99.85% identity in J2E3. J1E4 was detected to carry both the *qnS1* gene and mutations (gyrA p.S83L). The other three strains were only detected with mutations. Two mobile-colistin resistance genes *mcr-1.1* and *mcr-3.1* have been detected in strain J1E4 and J2E3, respectively.

### 3.3. Phenotype and Genotype Comparison

A brief comparison was done between the resistance characterization with phenotypic and sequencing methods. The phenotypic susceptibility was determined with the microdilution MIC test and disc diffusion according to the Clinical and Laboratory Standards Institute (CLSI) standards (Appendix A). According to the ESBL definition of the CLSI standards, all the isolates were determined with ESBL-producing phenotype based on the difference of MICs between subjects to cephalosporin and cephalosporin combined with clavulanic acid. The confirmation of ESBL-producing with MICs is agreed with the double-disc synergy test and genetic prediction (Table 3). Four out of nine isolates were confirmed as MDR with the phenotypic test. Five of the isolates were harboring resistance genes without showing the phenotype, which may due to the genes were regulated or silenced. 

All the isolates have shown resistance to 6/8 cephems tested, except that, no isolates were resistant to cefoxitin, but the MICs of J1E1, J1G1, and J2E3 have reached the intermediate range, as MIC = 16 µg/mL. The isolates have shown different susceptibility to ceftazidime and ceftazidime/clavulanic acid. Among the five isolates that harbored *bla*_CTX-M-15,_ four are resistant to ceftazidime, except J2E2 only reached the intermedium range. Besides, strain J2E3 is also resistant to ceftazidime, the *bla*_CTX-M-55_ it harbored is a variant of *bla*_CTX-M-15_. As a beta-lactamase inhibitor, clavulanic acid significantly influenced the susceptibility to ceftazidime and cefotaxime. The MIC of J1E4 to ceftazidime decreased 1024-fold when it was combined with the clavulanic acid. Except that, the resistance of J1E1 to cefotaxime was not inhibited by clavulanic acid, which is different from the response of the other four *bla*_CTX-M-15_ harbored isolates.

Among the 4 isolates which were carrying *qnrS1* genes, strain J1E4 and J2E4 have shown resistance to ciprofloxacin, but J2E2 and J2E3 were sensitive to ciprofloxacin. For the three isolates that were only detected with ciprofloxacin resistance mutations, J1E3 was sensitive to ciprofloxacin, the other two were resistant to ciprofloxacin.

Only the J1E4 strain has been determined to be resistant to colistin as MIC = 4 µg/mL, which may because of the *mcr-1* gene, besides, the mcr-1 gene is confirmed to be conjugatable and successfully provide resistance to the new receipts strain J53. Strain J2E3 is still sensitive to colistin even though harboring an *mcr-3* gene. 

### 3.4. Phylogenetic Analysis

Considering the clinic importance of *E. coli* ST131, a phylogenetic analysis was applied including the J2E1 ST131 and other ST131 clinical isolates from seven countries including Singapore as shown in Figure 1. If shown with the branch length, the branch containing J2E1 has a further evolution distance compared to other branches. All isolates on this branch including J2E1 were carrying *fimH41*, which may reflect that this branch belonged to ST131 clade A. The nearest strain to J2E1 on the tree is strain Mer-335 which has been isolated from the blood of a patient in Singapore in 2016. Besides this strain, 3 clinical isolates from other countries: Australia, New Zealand, and Indonesia were also on the same branch as J2E1. However, as the SNP difference was over 350, this may suggest a more distant common ancestor, instead of direct relationships between the clinical isolates and J2E1. 

Even though no direct link was found between the J2E1 and clinic isolates, the genetic analysis has shown some similarity of AMR markers including the similar types of beta-lactamase genes and other mutations among isolates on clade A branch (Appendix A). Four out of six isolates on this branch including J2E1 were carrying *bla*_CTX-M-27_, except Mer-277 and Mer-335, harboring *bla*_CTX-M-15_ and *bla*_CTX-M-14_, respectively. The *bla*_CTX-M-27_ of J2E1 and Mer-304 were both located on short contigs, below 2000 bp, with 100% identification. A conversed transposase DDE domain was found in front of the *bla*_CTX-M-27_ on the two contigs, which may suggest the *bla*_CTX-M-27_ was located on a widely distributed transferable element, shared between these two isolates. 

The *gyrA* p. S83L mutation point appeared in all 6 isolates of this clade, *parC* p.S80I and *gyrA* p. D87N mutations of J2E1 were also detected in Mer-304 and Mer-335. Both Mer-304 and J2E1 were harboring unique mutations on *parE* as p.L445H and pE460D, respectively; all the other 4 isolates on this branch were harboring *parE* p.I529L mutation instead, which suggested the common existence of quinolone related mutations in ESBL-producing isolates of ST131 clade A from both clinic and environment. 

### 3.5. Location Determined of Selected AMR Genes and Genetic Environment Analysis of Bla_ctx-Ms_

To predict the location of resistance genes with sequencing data, the contigs contain resistance genes were BLAST with BLASTn to find the best hits (Appendix A). Four of the nine contigs harboring *bla*_CTX-M_ genes were found highly similar to the chromosomes sequence, all four isolates were collected from sampling point 1. Further DNA alignment and annotation have investigated the similar genetic environment of the *bla*_CTX-M_ gene cluster (Table 4). The same type of insert sequence ISEcp1 was found in the forward direction of the *bla*_CTX-M_ gene, with a distance of either 48 or 42 bp (Figure 2). A *WbuC* family cupin fold metalloprotein was found that followed the *bla*_CTX-M-15_ in a reverse direction. Interestingly, we found this gene cluster highly converses, as also present on the other two contigs carry *bla*_CTX-M-15,_ which predicted to locate on the plasmid. This may reflect an important hypothesis that this gene cluster can jump from the plasmid to the chromosome. As the genes before and after this cluster are highly different, it is more likely this cluster is randomly inserted into the chromosome. 

All of the four *qnrS1* genes were found to be located on plasmids, and three of them were co-located on the same contigs with *bla*_CTX-M_ genes. Interestingly, the *qnrS1* gene in J2E2 was followed with another insert sequence ISKpn19; besides, the fragment between two inserts is the same as the sequence of the fragment from J2E4, which also contain ISEcp1, *bla*_CTX-M-15_, and *qnrS1*. This may raise the hypothesis that the fragment between two inserts potentially forms a co-transfer cluster. 

Both of the *mcr* genes were predicted to locate on the plasmid. No perfect hit was found for the contig carrying *mcr-1.1* in J1E4, the hit with the highest coverage (52%) found by Blastn was an IncFIA(HI1)/IncHI1A/IncHI1B(R27) type plasmid from *E. coli*. Only the first 7000 bp of the contig showed 100% identity. This prediction has been further proved with the conjugation test. The contig carrying *mcr-3.1* of strain J2E3 showed a high similarity to a plasmid of a sewage isolate from Japan, which may suggest the *mcr-3.1* of J2E3 is also located on a plasmid. 

### 3.6. Co-Conjugation of Mcr-1 Gene and Bla_tem-1_ Gene

The *mcr-1* gene was successfully conjugated to the J53 strain and confirmed with the PCR test. Interestingly, as tested with beta-lactamase genes specific primer on the transconjugants, a co-existing of the *bla*_TEM-1_ gene was found, but not the *bla*_CTX-M-15_ gene. The co-conjugation maybe because these two genes were both predicted to be located on plasmids, while the *bla*_CTX-M-15_ gene was predicted to be chromosome-located instead by Blastn. From this angle, the genetic prediction of location is agreed with the transferability phenotype. 

We also compared the antimicrobial susceptibility of transconjugants and the donor strain (Table 5). The colistin resistance phenotype is successfully presented in the transconjugants and even the MIC of colistin increased twice. Due to the *bla*_TEM-1B_ gene is also detected in the transconjugants, the transconjugant strain has also shown resistance to both the ampicillin and cephalosporin beta-lactams. However, the transconjugant strain is less responsive to clavulanic acid on ceftazidime resistance, which may be due to the missing *bla*_CTX-M-15_ gene. Even so, the transconjugant still showed the ESBL-producing phenotype. Interestingly, besides the resistance to beta-lactams, the transconjugant has also shown a higher MIC to ciprofloxacin, and a similar level of resistance to tetracycline and chloramphenicol, which suggest the co-conjugation of these resistances with *mcr-1*.

## 4. Discussion

An increasing prevalence of ESBL-producing bacteria has been reported from different sources including food, animal, and clinical isolates globally [18]. Besides the potential risk of transmitting ESBL-producing isolates from other sources to humans suggested [19], the spreading of ESBL genes through horizontal gene transfer (HGT) has also become a big concern to public health. Concerning this, more deep genetic studies are needed. As shown in this study, WGS has become a useful tool to study the ESBL-producing isolate from different sources. The phylogenetic analysis with WGS can help to draw the links between isolates from different sources, and further DNA alignment can help to predict the AMR gene cluster spread between different isolates. In Singapore, the WGS has been widely used for ESBL studies of isolates from human, food, and animals, but not for isolates from reservoir water [20]. One study has reported resistant bacteria from aquaculture farms in Singapore without WGS analysis [21]. To our knowledge, our study is the first WGS analysis report of reservoir ESBL-producing isolates from Singapore. 

The ESBL-producing *E. coli* has great clinical importance as a high prevalence of MDR, which limits the options for infection control [22]. The co-existence of ESBL genes and other resistance genes has been widely reported and was also shown in our isolates. All isolates were detected to carry more than 2 classes of resistance genes, and four of the isolates were resistant to both beta-lactams and ciprofloxacin. Besides, more than one type of Ori was detected in these isolates, which may suggest their high capabilities in gene communication. Moreover, six out of the nine isolates were carrying contigs with more than one type of AMR genes, which may suggest the co-transfer potential of these AMR genes. The most significant co-location in our isolates is the fluoroquinolone resistance genes and beta-lactamase. Fluoroquinolone and beta-lactams are both important and widely used antimicrobials in the clinic. The co-transfer of these two types of genes would be a potential risk to public health. 

*E. coli* ST131 has been recognized as one of the top contributors to urinary tract infections in the human clinic globally [4]. *E. coli* ST 131 has been reported to evolve with MDR especially the ESBL-producing phenotype quite often. Research in the United States has pointed out the ST131 as the major cause of serious MDR infections in 2007 [23]. The J2E1 (ST131 clade A) isolate was detected to carrying the virulence factor of *gad* (glutamate decarboxylase), *iha* (adherence protein), and *senB* (plasmid-encoded enterotoxin). The PathogenFinder has also suggested its significant potential as a human pathogen. J2E1 was also determined to be MDR, and resistance genes subjected to six antimicrobials were detected including a *bla*_CTX-M-27_. Unlike reported in other countries like Denmark, which have shown ST131 is highly associated with *bla*_CTX-M-15_ [24], *bla*_CTX-M-27_ has also been reported with a high prevalence among ST131 from bloodstream infections in other research in Singapore. Even though infections caused by clade C were dominant among ST131 *E. coli*, blood infections caused by clade A isolates carrying *bla*_CTX-M-27_ has also been reported in Singapore [25]. The isolate J2E1 collected here is O16: H5, a similar serotype that has been reported in some countries like France recently [26]. This serotype has also often been reported in clade A ST131, which is different from the abundant reports of O25: H4, mainly found in clade C. Even though no direct relationships were found between the water isolates and the isolates from the clinic, the same plasmid type, mutations, and similar resistance profiles were detected among them, and especially some conserved sequences contain the resistance genes were found, still suggesting a high potential horizontal AMR gene transfer between aquatic isolates and the clinical isolates. 

The chromosome located *bla*_CTX-M-15_ has been widely reported already [27]. The highly conserved structure has been found in many different species, as contained ISEcp1, following with resistance genes and a reversed *WbuC* family, cupin fold metalloprotein. The *WbuC* family protein likely contains a histidine-residue as a metal-binding ligand and has been reported to relate to O-antigen biosynthesis [28]. The *WbuC* gene at this special position was also frequently annotated as Tryptophan synthase, due to the similar sequence protein that has been reported in *Klebsiella* spp. However, as the functions of the protein in the *WbuC* family were highly varied in different species, the role of it here with *bla*_CTX-M-15_ is still not well documented. The resistance genes in this unique gene cluster are not limited to *bla*_CTX-M,_ and *mcr-9* has also been reported to be followed by *WbuC* genes recently [29]. Therefore, the functional studies of the *WbuC* genes will be very interesting in the future. Even though the five isolates with *bla*_CTX-M-15_ are carrying the same gene cluster, their MIC of different beta-lactams and their response to beta-lactamase inhibitor is still different. The regulation and cooperation of ESBL genes are still important topics in the future.

Two of the isolates were detected with mobile-colistin-resistance genes, as *mcr-1.1* and *mcr-3.1,* respectively, but only J1E4 with *mcr-1.1* has shown resistance to colistin. The mcr-type genes were firstly reported in 2015. Colistin is also known as polymyxin E and has been taken as the last resort to treat MDR infections; therefore, the transferable colistin resistance has been related to a high risk in the clinic. The *mcr-type* genes in *E. coli* isolates have been reported in Singapore from other sources like patients in hospitals [30] and food [31]. The mcr-type genes from environmental water sources have also been reported in other countries like China [32] and Switzerland [33]. However, to our knowledge, no genetic report of *mcr-type* genes in environmental water from Singapore has been published yet. The co-existence of *mcr-type* genes and ESBL genes have already been reported in other Asian countries like China [34]. However, based on the Blastn prediction, the *mcr-1* and *bla*_CTX-M-15_ genes are very unlikely to be co-located. Further conjugation studies failed to show their co-transferability, which has been suggested in other studies [35]. Instead, other types of resistance correspond with the colistin resistance including the *bla*_TEM-1B_ gene detected with PCR and shown with the phenotype test. This also reflects the HGT of a group of AMR genes instead of only one. 

The disagreement between genotype and phenotype is quite significant for fluoroquinolone resistance among our isolates as well as other resistance that have been shown in the phenotype and genotype comparison. Three out of seven isolates were sensitive to fluoroquinolone, disagreeing with the prediction by resistance genes and mutations. This phenomenon may be attributed to the “turn off” of the expression by transcriptional regulation or other gene silencing mechanisms. Further studies are needed to investigate the mechanism. However, non-resistant phenotype does not mean there is no risk at all. The *qnrS1* genes still have the potential to spread and meet a suitable host without the regulation to express. The site mutation is also important for the fluoroquinolone resistance as shown in the two isolates. Hopefully, with more WGS data update, the mutation database will increase to improve the accuracy of resistance detection. 

Even though no isolates were directly linked to clinical cases in this research, the same sequence type has been found in both reservoir isolates and clinical isolates, including ST131, ST38, and ST10. This may also be because the water collection period was one-two years later compared to clinic isolates collections. However, ST131 is highly evolved with humans and very unlikely to be the natural inhabited strain in water; this may suggest potential contamination from humans to the reservoir water. The core genome alignment has increased the accuracy of isolate tracking compared to only using MLSTs; therefore, it can show the detailed difference between isolates. So far there is quite limited information on AMR genes from the water source in Singapore, which also means there is a great need to apply NGS in future AMR studies of the aquatic environment. Theoretically, the separated water systems in Singapore should show advantages in avoiding waste contamination. However, as the sequencing data are limited, we cannot draw this conclusion from this research. However, in the future, with a more complete AMR surveillance system, we can further evaluate the benefit of this system and provide more scientifically-based suggestions to control the AMR spreading in reservoir water.

## 5. Conclusions

In a summary, this study has presented the existence of ESBL-producing *E. coli* in the surface water (reservoir water) in Singapore, including high-risk strain like ST131 isolate, and proved they are carrying conversed *bla*_CTX-M_ genes clusters present on both plasmid and chromosome, as well as conjugatable resistance genes as the *mcr-1*.

## Figures and Tables

**Figure 1 ijerph-18-00937-f001:**
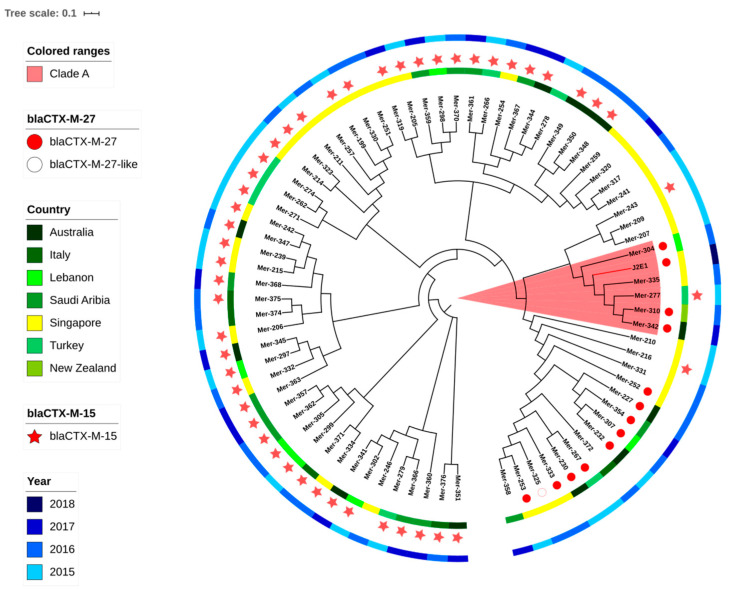
The phylogenetic tree of ST131 isolates from both the clinic and reservoir based on core genome single-nucleotide polymorphisms (SNPs). The tree was built with E. coli MG1655 as the reference genome and branch length was ignored. J2E1 was isolated in this research and or the other isolates were from NCBI. The metadata is shown in the Appendix A.

**Figure 2 ijerph-18-00937-f002:**
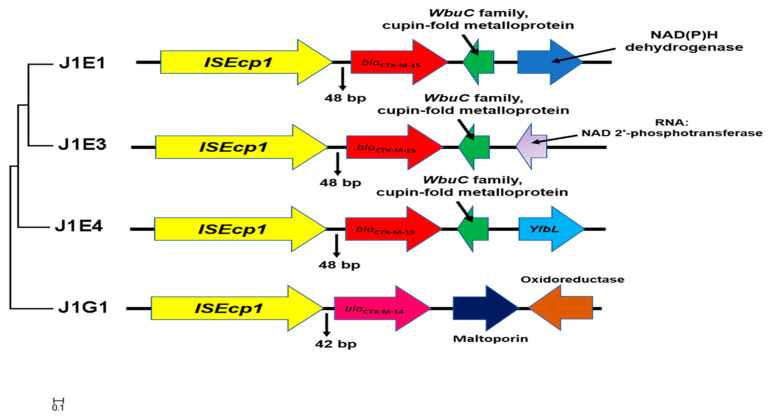
The genetic environment surrounding *bla*_CTX-M_ genes located on the chromosome. The maximum likelihood tree was built based on the alignment of 5000 bp fragments contain insert sequence and *bla*_CTX-M_ gene extract from the chromosome. The gene annotation and direction were generated by the RAST server and further corrected with NCBI blast. The conversed gene cluster contains *ISEcp1*, *bla*_CTX-M-15,_ and *WubC* family metalloprotein.

**Table 1 ijerph-18-00937-t001:** Basic information of the isolates.

Strain ID	GC Content/%	Genome Size/bp	Kmer Result	MLST	Plasmid Ori Detected
**J1E1**	50.6	5,111,241	*E. coli*	68	IncFII(pCoo) *, Col156 *
**J1E2**	50.5	4,953,866	*E. coli*	457	Incl1 *,ColpVC *
**J1E3**	50.3	5,176,493	*E. coli*	127	IncFII(pRSB107), IncFIB *,IncFIA,Col156 *
**J1E4**	50.5	4,866,444	*E. coli*	10	IncFIA(HI1),IncHl1A,IncHl1B(R27)
**J1G1**	50.3	5,408,697	*E. coli*	648	IncFII(pRSB107), IncFIB *,IncFIA *,Col156 *,Col(MG828) *
**J2E1**	50.7	5,006,788	*E. coli*	131	IncFII(pRSB107), IncFIB *,IncFIA,Col156 *
**J2E2**	50.6	5,088,348	*E. coli*	38	IncFII(pHN7A8) *,ColpVC *
**J2E3**	50.5	5,248,876	*E. coli*	1730	IncFIB *,IncFII,IncHl2,IncHl2A *
**J2E4**	50.6	4,886,227	*E. coli*	215	IncFIA(HI1) *,IncHl1 *,IncR *, IncX4, IncFIA *

* without 100% identification.

**Table 2 ijerph-18-00937-t002:** Acquired resistance genes and resistance-related mutations detected in isolates by ResFinder.

Isolates ID	Aminoglycoside	Beta-lactam	Colistin	Fluoroquinolone	Fosfomycin	MLS
J1E1		*bla* _CTX-M-15_			*fosA4*	*mdf(A) *,mph(A)*
J1E2		*bla* _CTX-M-8_				*mdf(A) **
J1E3	*aph(3″)-lb, aadA5, aac(3)-lld*, aph(6)-ld*	*bla*_TEM-1B_,*bla*_CTX-M-15_		*gyrA* p.S83L		*mdf(A) *,mph(A)*
J1E4	*aadA1**	*bla*_TEM-1B_,*bla*_CTX-M-15_	*mcr-*1.1	*qnrS1, gyrA* p.S83L		*mdf(A) **
J1G1		*bla* _CTX-M-14_		*gyrA* p.S83L, p.D87N, *parE* p.S458A*, parC* p.S80I		*mdf(A)**
J2E1	*aph(3′’)-lb, aadA5, aph(6)-ld*	*bla* _CTX-M-27_		*parC* p.S80I,*parE* p.E460D, *gyrA* p.D87N, *gyrA* p.S83L		*mdf(A) *,mph(A)*
J2E2		*bla* _CTX-M-15_		*qnrS1*		*mdf(A) **
J2E3	*aph(3′)-la *, aadA2, aac(3)-lld *, aph(6)-ld,aph(3′’)-lb,aadA1**	*bla* _CTX-M-55_	*mcr-*3.1	*qnrS1**		*mdf(A) *,mph(A)*
J2E4		*bla* _CTX-M-15_		*qnrS1*		*mdf(A)**
**Isolates ID**	**Phenicol**	**Sulphonamide**		**Tetracycline**	**Trimethoprim**	**Total resistant genes number**
J1E1						4
J1E2						2
J1E3		*sul1,sul2*		*tet(A)*	*dfrA17*	12
J1E4	*floR **	*sul3*		*tet(A)*		9
J1G1					*dfrA17*	3
J2E1		*sul1,sul2*		*tet(A)*	*dfrA17*	10
J2E2						3
J2E3	*floR *,catA2 **	*sul1,sul3*		*tet(A) **	*dfrA12*	17
J2E4						3

MLS: macrolide, lincosamide, and streptogramin B, *: identification below 100%.

**Table 3 ijerph-18-00937-t003:** The comparison of phenotypic and genotypic resistance.

Isolates ID	Aminoglycoside		Beta-lactam		Colistin		Fluoroquinolone	
	Genotype	Phenotype (Gentamic, MIC)	Genotype	Phenotype (MIC)	Genotype	Phenotype (MIC)	Genotype	Phenotype (Ciprofloxacin, MIC)
J1E1	-	S	+	R	-	S	-	S
J1E2	-	S	+	R	-	S	-	S
J1E3	+	R	+	R	-	S	+	S
J1E4	+	S	+	R	+	R	+	R
J1G1	-	S	+	R	-	S	+	R
J2E1	+	S	+	R	-	S	+	R
J2E2	-	S	+	R	-	S	+	S
J2E3	+	S	+	R	+	S	+	S
J2E4	-	S	+	R	-	S	+	R
**Isolates ID**	**Phenicol**		**Sulphonamide**		**Tetracycline**			
	**Genotype**	**Phenotype (Chloramphenicol, Disc diffusion)**	**Genotype**	**Phenotype (Trimethoprim-sulfamethoxazole, Disc Diffusion)**	**Genotype**	**Phenotype (Disc Diffusion)**		
J1E1	-	S	-	R	-	S		
J1E2	-	S	-	S	-	S		
J1E3	-	S	+	R	+	R		
J1E4	+	R	+	S	+	R		
J1G1	-	S	-	S	-	S		
J2E1	-	S	+	R	+	R		
J2E2	-	S	-	S	-	S		
J2E3	+	R	+	R	+	R		
J2E4	-	S	-	S	-	S		

+/-: with/without resistance genes or related mutations, S: sensitive, R: resistance, Grey highlight: disagreement between genotype and phenotype.

**Table 4 ijerph-18-00937-t004:** Insert sequence and their location detected by ISFinder.

Isolate ID	Contigs Information		Resistance Information		Insert Sequence Information				
	Contigs ID	Contigs Length/Bp	AMR Genes	Position on the Contigs	Insert Sequence ID	Insert Length/Bp	Insert Position	IS Family	Accession
J1E1	27	269,951	*bla_CTX-M-15_*	197172..198047	ISEcp1	1656	195468..197123	IS1380	AJ242809
J1E3	151	128,211	*bla_CTX-M-15_*	86273..87148	ISEcp1	1656	84569..86224	IS1380	AJ242809
J1E4	28	128,728	*bla_CTX-M-15_*	43925..44800	ISEcp1	1656	42221..43876	IS1380	AJ242809
J1G1	80	110,660	*bla_CTX-M-14_*	83884..84759	ISEcp1	1656	82186..83841	IS1380	AJ242809
J2E2	31	289,403	qnrS1	242382..243038	ISKpn19 *	2851	238783..241633	ISKra4	NC_010886
			*bla_CTX-M-15_*	247679..248554	ISEcp1 *	1656	248603..250258	IS1380	AJ242809
J2E4	220	716,717	qnrS1	555846..556502	ISEcp1	1656	562067..563722	IS1380	AJ242809
			*bla_CTX-M-15_*	561143..562018					

* identification below 100%.

**Table 5 ijerph-18-00937-t005:** The minimum inhibitory concentration (MIC) comparison among J1E4, J53, and the transconjugants.

Antimicrobials	J1E4 (Donor)	J53Coli (Transconjugates)	J53 (Recept)
**Ceftriaxone**	>128	>128	<1
**Meropenem**	<1	<1	<1
**Cephalothin**	>16	>16	>16
**Cefpodoxime**	>32	>32	2
**Ciprofloxacin**	2	>2	<1
**Cefotaxime**	>64	>64	<0.25
**Gentamicin**	<4	<4	<4
**Cefotaxime/clavulanic acid**	<0.12\4	<0.12\4	<0.12\4
**Ampicillin**	>16	>16	<8
**Ceftazidime**	>128	32	1
**Cefazolin**	>16	>16	<8
**Ceftazidime/clavulanic acid**	0.25\4	0.5\4	0.5\4
**Imipenem**	<0.5	1	1
**Piperacillin/tazobactam constant 4**	>64\4	<4\4	<4\4
**Cefepime**	>16	>16	<1
**Colistin**	4	8	<0.25
**Cefoxitin**	<4	8	<4

The MIC was determined with the microdilution methods. The successful transconjugants gain MDR from the donor strain J1E4 including the resistance of colistin, 3^rd^ generation cephalosporins, and ciprofloxacin.

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
