# Peer review of "Characterization of Extended-Spectrum Beta-Lactamase-Producing Escherichia coli Isolates from Jurong Lake, Singapore with Whole-Genome-Sequencing"

_ijerph, 2021, doi:10.3390/ijerph18030937_

Round 1

Reviewer 1 Report

Authors have collected a water sample from a reservoir from Singapore and tested for the presence of ESBL producing E. coli harbouring blaCTX-M gene and other resistance markers and looked at using both genotypic and phenotypic methods including conventional MIC and disk diffusion method.

The title can be condensed as ' Characterisation of Extended -Spectrum Beta-Lactamase-producing Escherichia coli from Jurong Lake, Singapore.
The abstract may need some insights about their work/finding.

It would be useful if authors could add an image of the double-disk synergy test image with positive control in the manuscript.

I would like to see the reference point (reference value/cutoff) of resistant and sensitive determination shown in table 3.

Table 4 can be moved to the section of the Supplementary material.
The discussion is too long, authors could consider revising and reduce a bit of word.

Overall: This manuscript provides more insights into 'how the resistance determinants spread across the environmental/ clinical strains.
The positive isolates were extensively characterised.

Author Response

1) The title has been changed to "Characterisation of Extended -Spectrum Beta-Lactamase-producing Escherichia coli from Jurong Lake, Singapore with Whole Genome Sequencing. We want to keep the WGS in the title to address the novelty with sequencing analysis.

2) Double-disk synergy test image added as Fig S3 in supplementary.

3) Add the breakpoint of MIC tests and disc diffusion in the last row of Table S1 and Table S5 respectively. 

4)Move Table 4 to supplementary as Table S4.

5)Combine result 3.5 with 3.6 and revise to a shorter discussion. 

Reviewer 2 Report

1. In line 67, please leave the appropriate spacing between the words. Words spacing must be fixed in other places too, such as line 121.

2. In line 77- 80, authors claimed that “However, the reservoir …………human beings.” Could you please provide the appropriate citation to your claim.

3. Authors are encouraged to write why this research study is inevitable. In other words, they are also encouraged to justify the importance of this research in depth where previous research studies (if have any) could not fulfill.

4. In line 91, are those samples chosen randomly? Specify.

5. Please fix the citations throughout the paper. Do not give the link in the body.

Author Response

1) Words spacing problem may due to the Justify format mode used for MSwork. I tried to avoid it.

2) provide citation accordingly. 

3) added at the beginning of the discussion. 

4) sampling point chose randomly, specified in methods accordingly. 

5) citations added already. And the links added are to convenience the reader who may reference this paper to use it easier. The links also commonly appear in other publications like the samples below:

Aung, Kyaw Thu, et al. "Salmonella in Retail Food and Wild Birds in Singapore—Prevalence, Antimicrobial Resistance, and Sequence Types." International journal of environmental research and public health 16.21 (2019): 4235.

Guo, Siyao, et al. "Prevalence and genomic analysis of ESBL-producing Escherichia coli in retail raw meats in Singapore." Journal of Antimicrobial Chemotherapy (2020).

But can be removed if necessary. 

Reviewer 3 Report

The Authors wrote an article on the characterization of clinically relevant 1antimicrobial resistance genes with whole-genome sequencing in Extended-Spectrum  Beta-Lactamase-Producing Escherichia coli isolates from Jurong Lake, Singapore. Despite reporting potentially interesting findings the manuscript presents several issues that should be addressed in order to improve its quality.

Major concerns

  • The use of the English language requires extensive revision. In the current form results and conclusions sections are not easily understandable.
  • The Results and Discussion sections are dispersive and could be significantly shortened to report and comment the most significant data (for example the characterization of the genetic context of just ESBL and mcr genes).
  • Table 4 should report the accession number of best hit records.

Author Response

1) The result and discussion part has been revised.

2) Combine the results 3.5 and 3.6 to shorten the result to make it clearer and highlight the main points (ESBL and mcr). Shorten the discussion part addressed the most significant findings. 

3) Table 4 has been moved to supplementary as Table S4, accession number is in the last row of the table.